# Evaluation of infrared thermography as a non-invasive method of measuring the autonomic nervous response in sheep

**Mhairi A. Sutherland**[¤a©]*, **Gemma M. Worth**[¤a©], **Suzanne K. Dowling, Gemma L. Lowe, Vanessa M. Cave, Mairi Stewart**[¤b©]

AgResearch Ltd., Ruakura Research Centre, Hamilton, New Zealand

© These authors contributed equally to this work.
¤a Current address: LIC, Hamilton, New Zealand
¤b Current address: Ministry for Primary Industries, Wellington, New Zealand
* Mhairi.Sutherland@agresearch.co.nz

**Data Availability Statement:** All relevant data are within the manuscript and its Supporting Information files.

## Abstract

Eye temperature measured using infrared thermography (IRT) can be used as a non-invasive measure of autonomic nervous system (ANS) activity in cattle. The objective of this study was to evaluate if changes in eye temperature (measured using IRT) can be used to non-invasively measure ANS activity in sheep. Twenty, 2 to 4-year-old, Romney ewes were randomly assigned to receive either epinephrine (EPI) or physiological saline (SAL) for 5 min administered via jugular catheter (n = 10 ewes/treatment). Eye temperature (˚C) was recorded continuously using IRT for approximately 25 min before and 20 min after the start of infusion. Heart rate and heart rate variability, measured using the root mean square of successive differences (RMSSD) and the standard deviation of all inter-beat intervals (SDNN), were recorded for 5 min before and up to 10 min after the start of infusion. Blood samples were taken before and after the infusion period to measure plasma epinephrine, norepinephrine, cortisol and packed cell volume (PCV) concentrations. During the infusion period, maximum eye temperature was on average higher (P<0.05) in sheep that received epinephrine than those that received saline. On average, heart rate was higher (SAL: 87.5 beats/min, EPI: 123.2 beats/min, SED = 7.07 beats/min; P<0.05), and RMSSD (SAL: 55.3 ms, EPI: 17.3 ms, SED = 14.18 ms) and SDNN (SAL: 54.3 ms, EPI: 21.5 ms, SED = 10.00 ms) lower (P<0.05) in ewes during the 5 min post-infusion period compared with ewes that received saline. An infusion of epinephrine resulted in higher geometric mean epinephrine (P<0.05) and cortisol (P<0.05) but not norepinephrine (P>0.05) concentrations in ewes compared to an infusion of saline. PCV concentrations were higher (P<0.001) by 7 ± 1.0% (mean±SED) in ewes after an epinephrine infusion. These results suggest that heart rate variability is a sensitive, non-invasive method that can be used to measure ANS activity in sheep, whereas change in eye temperature measured using IRT is a less sensitive method.

**Funding:** This work was funded by the New Zealand Ministry of Business, Innovation and Employment (formerly the New Zealand Foundation for Research, Science and Technology; Contract C10X0813; Wellington, New Zealand). MBIE and AgResearch had no role in the study design, data collection and analysis, interpretation of data, preparation of the manuscript or decision to publish. AgResearch Ltd. (Hamilton, New Zealand) provided support in the form of salaries for authors [MAS, GMW, SKD, GLL, VMC, MS] and research materials. The specific roles of these authors are articulated in the 'author contributions' section.

**Competing interests:** The authors are commercially affiliated with the New Zealand Ministry of Business, Innovation and Employment (MBIE; Wellington, New Zealand) and AgResearch Ltd. (Hamilton, New Zealand). AgResearch Ltd. (Hamilton, New Zealand) provided support in the form of salaries for authors [MAS, GMW, SKD, GLL, VMC, MS] and research materials. This does not alter our adherence to PLOS ONE policies on sharing data and materials.

## Introduction

Animals respond to stressors both behaviourally and physiologically, which enable them to adapt to these challenges; these changes can be measured and used to assess an animal's welfare. Physiological responses to stress include activation of the autonomic nervous system (ANS) and hypothalamic-pituitary adrenal axis. Activation of the ANS is the initial response to stress, also known as the fight/flight response, and provides more information about acute stress and the fear response than the slower responding hypothalamic-pituitary adrenal axis [1]. The ANS can be measured in farm animals based on changes in catecholamine concentrations, heart rate, heart rate variability (HRV), and stress-induced hyperthermia [1]. Catecholamines (e.g., epinephrine and norepinephrine concentrations) can be challenging to measure due to their short half-life in plasma and the need to handle animals in order to collect blood samples, which in itself can stimulate the release of catecholamines. Heart rate reflects both the sympathetic and parasympathetic activity of the nervous systems, whereas HRV provides information on the balance between the activity of the sympathetic and parasympathetic branches of the ANS [2]. Though heart rate and HRV are useful measures of ANS activity in animals, their measurement is reliant on animals being fitted with monitoring devices (e.g., heart rate monitors); systems that require animals to carry equipment could potentially cause changes in physiology and behaviour [3]. Changes in core body and surface temperatures can also be used as an indicator of acute stress in animals [4] and these changes in temperature can be measured non-invasively using infrared thermography (IRT). Therefore, IRT can potentially be used as a non-invasive method to measure stress in animals [5].

Infrared thermography is a non-invasive method of detecting the amount of infrared energy an object radiates [6] and based on a change in peripheral blood flow [7] can be used to non-invasively measure changes in an animal's surface temperature [5]. Infrared thermography has been used to detect changes in surface temperature across a range of anatomical regions, and over a range of different species (as reviewed by Cook and Schaefer [5]). Temperature of the eye region has commonly been measured in many species because this area is not affected by the presence of hair [7]. Additionally, the lacrimal caruncle region of the eye has a rich capillary bed innervated by the sympathetic nervous system [8]. Therefore, the eye offers itself as an ideal location for measuring changes in blood flow relating to the ANS [7]. Changes in eye temperature, measured using IRT, have been shown in response to a range of stressors in a variety of species [8–16].

Throughout their lifetime sheep may be exposed to different types of stressors. These stressors can be both physical and psychological, and stem from factors such as inclement weather, malnutrition, handling, painful husbandry procedures and transport [17]. As with other species, there are few non-invasive measures of the ANS available to researchers for use in sheep. Previous research suggests that IRT could potentially be used to measure stress in sheep: lacrimal caruncle temperature was higher during restraint and a voluntary approach test in sheep [15], however, a 30 min noxious ischaemic (painful) stimulus caused no changes in eye temperature in sheep [18]. Due to the inconsistent stress related changes in eye temperature reported in sheep to date, we wanted to validate whether eye temperature (measured using IRT) could be used to assess ANS activity in sheep. To achieve this, we used epinephrine to stimulate an increase in the ANS response in sheep based on methodology described in Stewart et al [19]. An epinephrine infusion was used rather than a biological stressor, so as not to confound the physiological with the psychological aspects of the stress response. The objectives of our study were to first, determine whether stimulation of the ANS response using epinephrine would cause changes in eye temperature in sheep and second, compare these changes in eye temperature with validated measures of ANS activity (e.g., heart rate, HRV, packed cell volume).

## Materials and methods

The study was undertaken in April 2012 at the Ruakura Research Centre, located in Hamilton, Waikato region, New Zealand. All procedures involving animals were approved by the Ruakura Animal Ethics Committee (protocol no. 12594) under the New Zealand Animal Welfare Act 1999.

### Animals and treatments

Twenty, 2 to 4-year-old Romney ewes (68.6 ± 5 kg, mean ± SD), were used in this study. Prior to the start of the study, ewes were managed outdoors on pasture under normal farm practice. Two weeks prior to the start of the study, ewes were moved into the research facility and acclimated to intensive handling and the facilities where the procedures would take place. During this time ewes were also acclimatised to restraint in a modified calf chute with a head catch (Cattlemaster calf bail, Te Pari Products, Oamaru, New Zealand), the IRT camera and to wearing heart rate monitors. At the end of the 2-week acclimation period, all animals appeared habituated to the research facility and the restraint chute, however no data was collected to confirm this. The trial took place over 5 days, with four ewes sampled each day (two ewes per treatment), one at a time.

One day prior to sampling, ewes were shorn down the left side of the body around the girth area to allow maximum contact for the heart rate monitors and down both sides of the neck to facilitate insertion of the jugular catheters. Bilateral indwelling jugular catheters were inserted, one for infusion and one for blood sampling. The catheters were removed immediately after sampling the following day.

Ewes were randomly assigned to one of two treatments (n = 10 ewes/treatment): epinephrine HCL (EPI: 6 µg/kg per min; Cat. No. E4642, Sigma Aldrich, St Louis, MO) dissolved in approximately 60 mL of physiological saline (SAL: Baxter Healthcare, New South Wales, Australia) or an equivalent volume of physiological saline. Infusions were delivered using a Baxter Flo-Gard 6201 volumetric infusion pump (Baxter Healthcare, Deerfield, IL). Ewes were given 1 h in a holding pen to settle before sampling started. Approximately 10 min before sampling started, each ewe was restrained manually, fitted with a heart rate monitor and then moved into the chute where it was restrained for infusion and sampling. Baseline data was collected for approximately 25 min followed by the infusion period (average duration: 05:50 ± 00:04 mins), after which post-treatment responses were monitored for a further 15 min. Over this period, cardiac activity and eye temperature was recorded and blood samples were collected (see below for actual timing of sampling and recording periods for each measure).

### Infrared thermography

Using an infrared camera (ThermaCam S60, FLIR Systems AB, Danderyd, Sweden), continuous infrared recordings were collected at a rate ~60 frames/s and a resolution of 640 x 480 pixels (S1 Movie). This was achieved through the use of image analysis interface software (ThermaCam Researcher, version 2.10, FLIR Systems AB, Danderyd, Sweden) which connected the camera to a laptop. Whilst restrained in the chute, infrared recordings of the eye region were collected from the left side of the ewe with the infrared camera positioned on a tripod at a set distance of 1 m and at a 90˚ angle from the direction the ewe was facing. The camera was set to an emissivity of ε = 0.98, based on the generally accepted emissivity of an animal's body [20] and as used previously in sheep [15, 21]. Ambient temperature (˚C) and relative humidity (%) were recorded in close proximity to the chute in which sheep were held every hour using a handheld logger (Kestrel 3000, Nielsen-Kellerman, Boothwyn, PA) and entered into the infrared camera to adjust for changes in atmospheric conditions. Infrared

recordings of the eye region were collected continuously for approximately 25 min before until 20 min after the start of the infusion period. From the continuous recordings, image analysis software (ThermaCam Researcher, version 2.7, FLIR Systems AB, Danderyd, Sweden) was used to select individual images (frames) comprising the recordings at 20 s intervals. Only images which were in focus and in which the eye was fully open were used as images which do not meet these requirements can affect the accuracy of the results. Maximum eye temperature (°C) was determined from each of these selected images by tracing a circle around the eye region as to include the medial, posterior palpebral border of the lower eyelid and the lacrimal caruncle (Fig 1).

## Heart rate

The inter-beat interval (IBI) was recorded every second using Polar heart rate monitors (RS800CX™, Polar Electro Oy, Helsinki, Finland), and heart rate and HRV data were analysed over the 5 min period prior to and up to 10 min after the start of the infusion period. Monitors were strapped firmly around the thorax immediately behind the forelimbs, with electrode contact points on areas from which wool had been removed and ultrasound transmission gel (Aquasonic 100, Parker Laboratories Inc., Fairfield, NJ) applied. At the end of each sampling period the stored heart rate was downloaded via a serial interface to a computer for analysis. The heart rate and IBI data were extracted using Polar ProTrainer 5 (version 5.35, Polar Electro Oy, Helsinki, Finland). Heart rate (recorded every second) was averaged over 1-min intervals, and IBI data were used to calculate HRV parameters, the standard deviation of all interbeat intervals (SDNN) and the root mean square of successive differences (RMSSD). For SDNN and RMSSD analysis, short segments of data containing 256 beats were examined for 5 min before and up to 10 min after the start of the infusion period. Before analysis, a correction

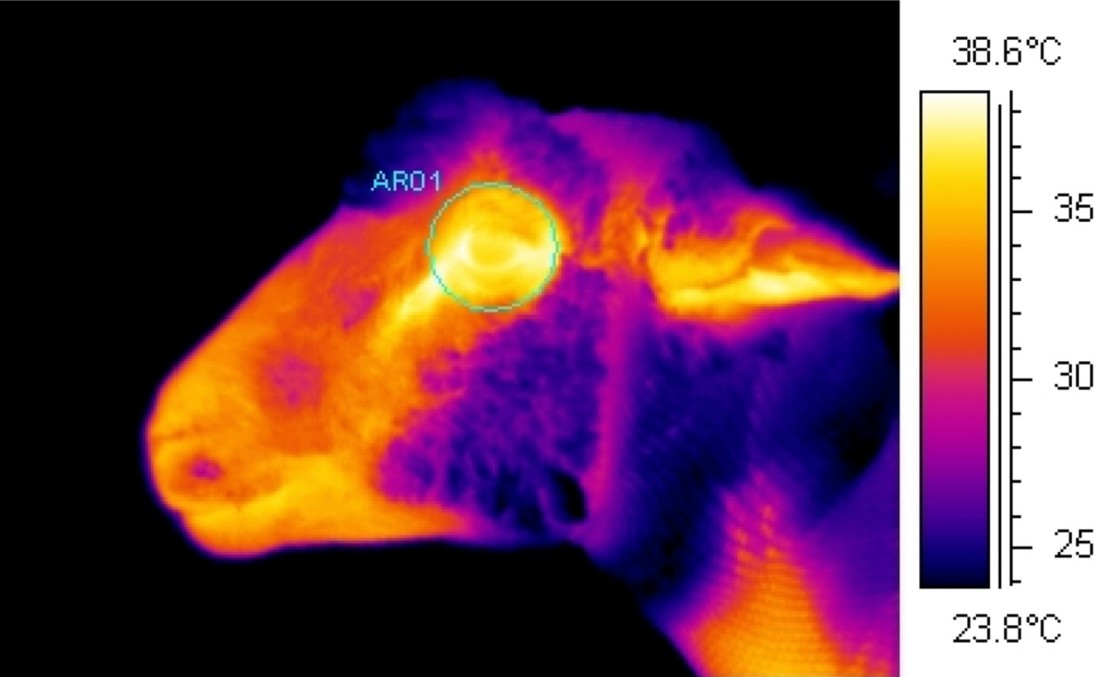

**Fig 1. Example infrared image.** Image shows the area traced around the eye region including the medial, posterior palpebral border of the lower eyelid and the lacrimal caruncle during analysis to determine the maximum eye temperature (°C).

function within the Polar software (Polar Precision Performance Software, version 4.03, Polar Electro Oy, Helsinki, Finland), set on default parameters, was used to correct for any artefacts and only data with an error rate of less than 5% were included in the analysis.

## Blood chemistry

To measure epinephrine and norepinephrine concentrations, continuous blood samples (6 mL) were collected at 30 s intervals into pre-chilled EDTA tubes (Becton Dickinson, Plymouth, UK) using a peristaltic pump starting -2 min and ending 10 min relative to the start of the infusion period (time 0). Additional blood samples (6 mL) were collected into tubes containing lithium heparin (Becton Dickinson) to measure cortisol and pack cell volume (PCV) concentrations. The maximum amount of blood taken per sheep was 250 mL. Blood samples were immediately placed on ice and centrifuged (2500 rpm) at 4°C for 10 min, within 10 min of collection. Plasma was stored at –80°C until analysed. Samples collected at –2, 0, 2, 4, 6, 8 and 10 min were assayed for epinephrine and norepinephrine concentrations using methodology described in Stewart et al. [19]. Briefly, 1 mL of plasma was extracted on alumina and the extracted catecholamines were separated and measured using reverse-phase HPLC. Cortisol concentrations were measured at –10, –5 and 15 min using a commercially available antibody radioimmunoassay kit (Coat-A-Count® Cortisol, Siemens; Los Angeles, CA). Packed cell volume was measured in samples taken at -5 and 5 min, using an ADVIA 120 Haematology system (Siemens Healthcare GmbH, Erlangen, Germany).

## Statistical analysis

The Bayesian mixed model smoothing programme, Flexi 5.60 (Upsdell, [22]), was used to determine the relationship between maximum eye temperature and treatment over time. The model forced the two treatment groups (epinephrine and saline) to be the same at baseline but allowed them to vary during the infusion and post-infusion periods. Individual animal differences and correlations between measurements taken on the same animal over time were allowed for. Fisher's least significant differences at the 5% level were used to compare the smoothed curves for the epinephrine and saline treatments over time.

For HR and HRV parameters, due to equipment malfunction, data were only collected from 10 ewes that received epinephrine and eight that received saline. For each animal, average HR, RMSSD and SNDD was calculated during infusion and over the 5-min periods before (baseline) and after infusion (post-infusion). These were analysed by repeated measures formulated as a linear mixed model and fitted using Residual Maximum Likelihood (REML). Correlations between measurements taken on the same animal over time were modelled using a power model of order 1. The significance of the treatment by time interaction (treatment x time) was assessed using an F-test and unprotected Fisher's least significant differences at the 5% level were used to compare the treatment means at each time period.

The epinephrine, norepinephrine and cortisol concentration data were similarly analysed, however prior to analysis, the concentrations were log-transformed to stabilise the variance. Furthermore, orthogonal contrasts were used to compare the mean change in log concentration following infusion between treatments.

Mean change in PCV (post-pre) between the two treatment groups were compared using one-way ANOVA. For each treatment group, a t-test was used to assess whether mean change was significantly different from zero.

Unless otherwise stated, all statistical analyses were performed using Genstat 19 (VSN International [23]). Residual diagnostic plots were inspected for evidence of departures from the assumptions of normality and homogeneity of variance.

## Results

During the infusion period, maximum eye temperature was higher on average ($P < 0.05$) in sheep that received epinephrine than those that received saline (Fig 2). As mentioned, due to the rich capillary bed located at the lacrimal caruncle, the maximum eye temperatures were typically recorded at this location within the eye region. Mean (± SD) maximum eye temperature before, during and after infusion: Baseline: SAL: 38.9 ± 0.17˚C, EPI: 38.9 ± 0.33˚C; Infusion: SAL: 38.9 ± 0.19˚C, EPI: 39.0 ± 0.42˚C; Post-infusion: SAL: 38.8 ± 0.20˚C, EPI: 38.8 ± 0.50˚C. Range of maximum eye temperature before, during and after infusion: Baseline: SAL: 38.4–39.4˚C, EPI: 38.0–39.4˚C; Infusion: SAL: 38.4–39.2˚C, EPI: 37.9–39.9˚C; Post-infusion: SAL: 38.4–39.2˚C, EPI: 37.8–40.3˚C.

There was no statistical evidence that heart rate differed between sheep that received epinephrine or saline during the infusion period, but during the 5 min post-infusion period, heart rate was on average higher ($P < 0.05$) in sheep that received epinephrine than those that received saline ($F_{2,33.5} = 27.98$, treatment x time: $P < 0.001$; Fig 3A). There was a tendency for a treatment x time effect on RMSSD ($F_{2,33.3} = 2.65$, $P = 0.085$; Fig 3B) and a statistically significant treatment x time effect on SDNN ($F_{2,33.8} = 3.41$, $P = 0.045$; Fig 3C). During the infusion period, average RMSSD and SDNN was not significantly different ($P > 0.05$) between saline and epinephrine infused sheep. During the post-infusion period, RMSSD and SDNN were on

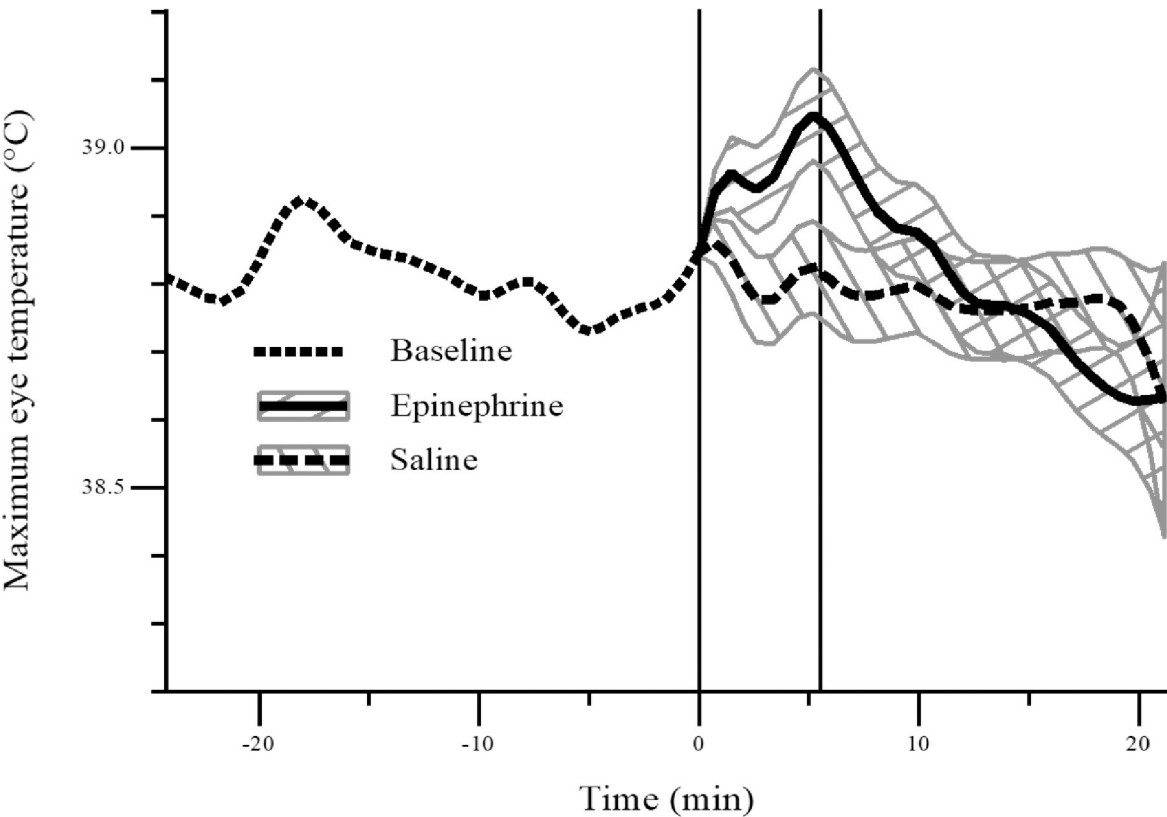

**Fig 2. Average maximum eye temperature (˚C) in sheep following an infusion (0 to 5 min) of epinephrine (n = 10) or saline (n = 10).** The plot gives the estimated smooth curve at baseline (dotted), and for the epinephrine (solid) and saline (dashed) treatment groups post-baseline. The grey bands are Fisher's least significant differences between treatments. The two treatments are significantly different at the 5% level where the bands do not overlap. The left-hand and right-hand vertical lines denote the start and end of the infusion period, respectively.

A

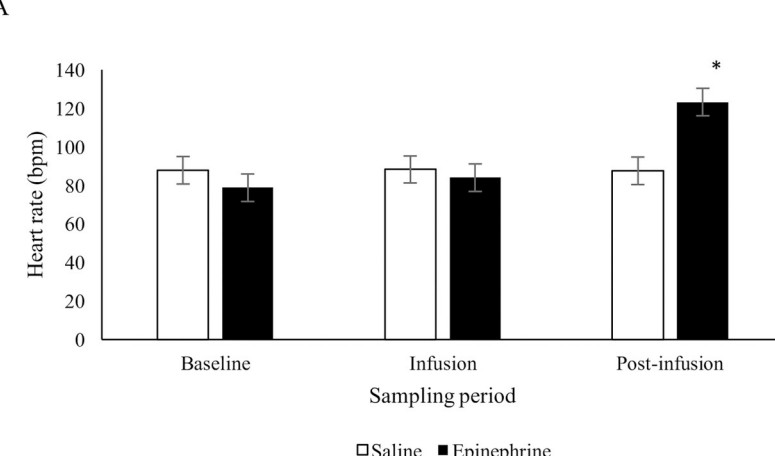

B

C

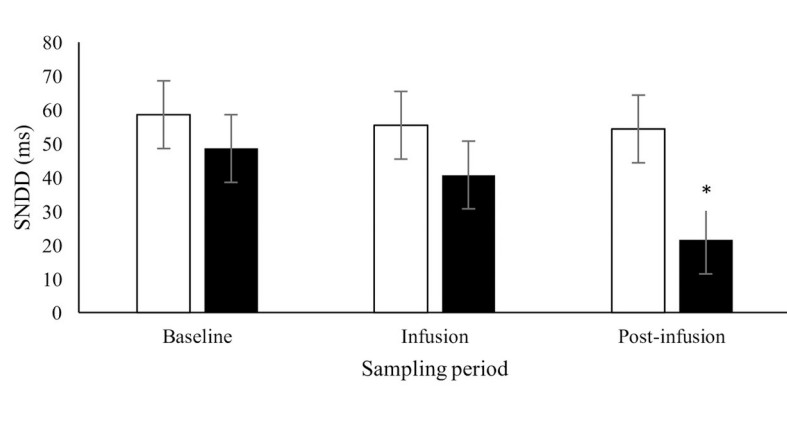

**Fig 3.** Mean (± SED) heart rate (bpm) (A), RMSSD (ms) (B) and SDNN (ms) (C) in sheep during a 5 min period before (baseline), during (infusion) and following (post-infusion) an infusion with epinephrine (n = 10) or saline (n = 8). *Treatments differ at $P < 0.05$.

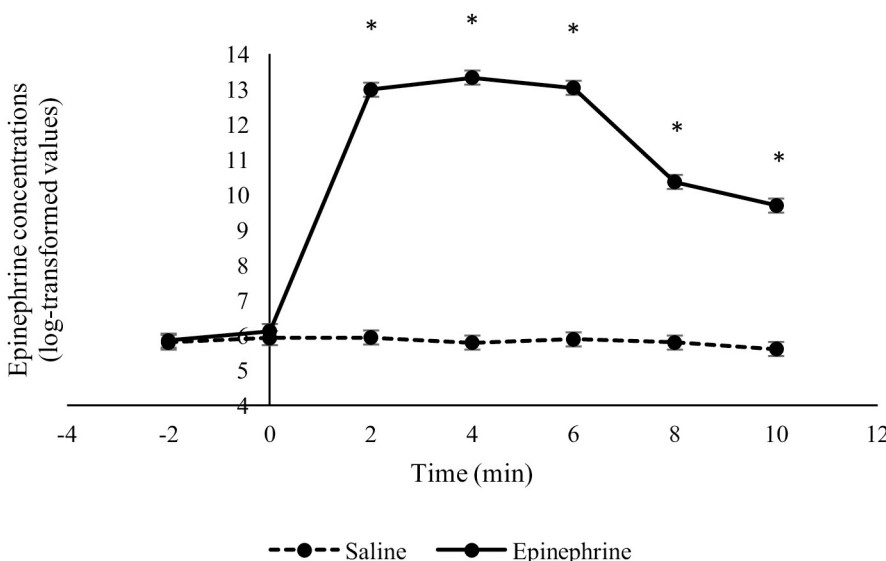

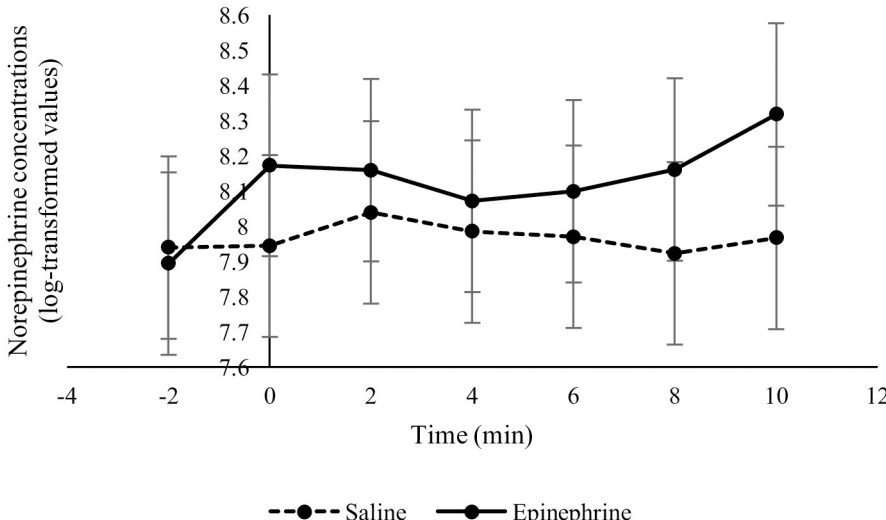

**Fig 4.** Epinephrine (A) and norepinephrine (B) concentrations (log-transformed means ± SED) in sheep in response to an infusion of epinephrine or saline (n = 10 / treatment). The infusion period started at time 0 and ended after 5 minutes. *Treatments differ at P < 0.05.

average lower ($P < 0.05$) in sheep that received epinephrine than those that received saline. There was no evidence of a difference in baseline values for HR, RMSSD and SDNN between sheep that received epinephrine or saline.

Overall, the epinephrine infusion affected epinephrine concentrations ($F_{2,94.5} = 390.7$, treatment x time: $P < 0.001$) but not norepinephrine concentrations ($F_{2,95.8} = 0.9$, treatment x time: $P = 0.508$) (Fig 4). Geometric mean epinephrine concentrations were higher (P < 0.05)

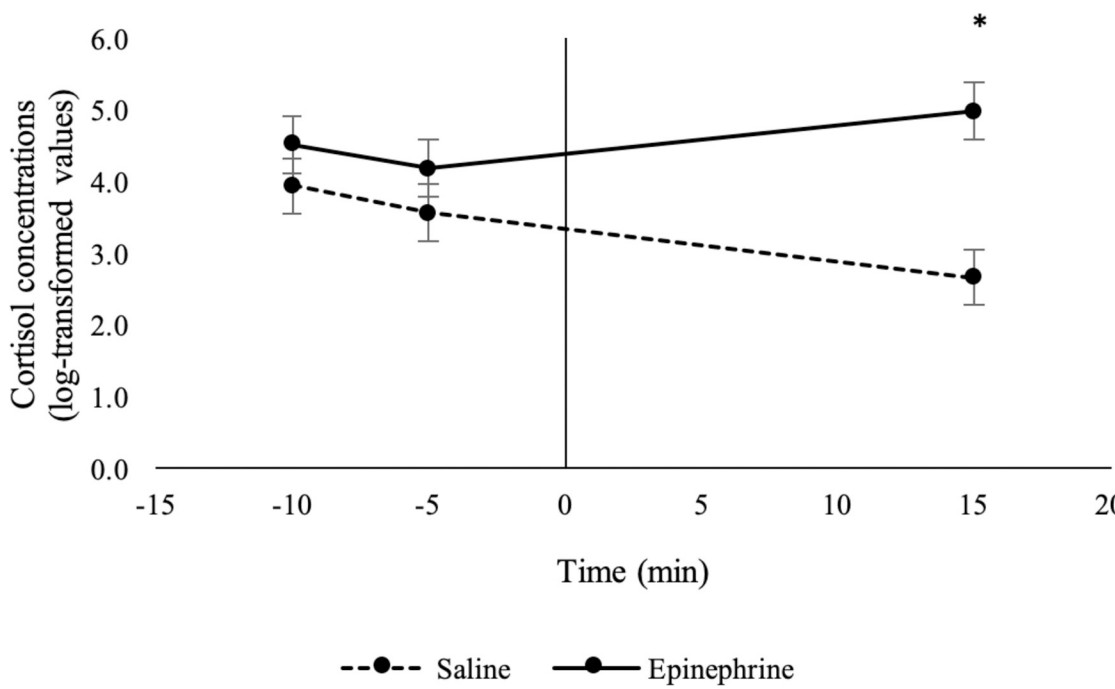

**Fig 5. Cortisol concentrations (log-transformed means ± SED) in sheep in response to an infusion of epinephrine or saline (n = 10 / treatment).** The infusion period started at time 0 and ended after 5 minutes. *Treatments differ at P < 0.05.

in sheep 2, 4, 6, 8 and 10 min after receiving an infusion of epinephrine compared to sheep that received an infusion of saline. Geometric mean norepinephrine concentrations were similar (P > 0.05) between sheep that received epinephrine or saline at all time points during the infusion and post-infusion periods.

Epinephrine infusion affected cortisol concentrations in sheep ($F_{2,31.3}$ = 21.1, treatment x time: $P < 0.001$). Geometric mean cortisol concentrations were higher ($P < 0.05$) in sheep that received epinephrine than those that received saline 15 min after the start of the infusion period (Fig 5). Cortisol concentrations did not differ ($P > 0.05$) between sheep that received epinephrine or saline at 10 or 5 min prior to the start of the infusion period (baseline period).

The PCV increased in sheep that received an epinephrine infusion but decreased in sheep that were given saline (mean change in PCV: epinephrine: 7%, saline: -1%, SED = 1.0%; $F_{1,16}$ = 58.3, treatment: $P < 0.001$).

## Discussion

Eye temperature, heart rate and HRV, and plasma epinephrine, cortisol and PCV concentrations all changed in response to an epinephrine infusion in sheep in the present study. Interestingly, eye temperature increased in the present study but decreased in cattle in response to a similar 5-min epinephrine challenge [19]. Acute psychological stress stimulates sympathetic-mediated cutaneous vasoconstriction, causing a rapid drop in skin temperature. This drop is then followed by a rise in core body temperature and a subsequent rise in peripheral temperatures as vasodilation occurs to dissipate excess heat [14]. This characteristic decrease in surface temperature and increase in core body temperature following exposure to a psychological stressor is termed stress-induced hyperthermia [4]. Numerous studies measuring eye temperature in response to stress show this characteristic response; eye temperature drops rapidly in response to a stressor and then increases back to baseline levels, sometimes increasing above

baseline levels. For example, this pattern was observed in cattle in response to an epinephrine challenge [19], disbudding [9], startle and an electric prod [12] and in chickens in response to restraint [13]. Conversely, eye temperature has been found to increase in response to castration [11] and catheterisation [8] in cattle, clipping [16] and noseband tightening [24] in horses, restraint in chickens [14] and sheep [15] and in response to a veterinary examination in dogs [25]. It is unclear why eye temperature sometimes increases in response to stress; however, it is hard to directly compare these studies due to differences in timing of data collection. For example, in some studies, IRT images were only taken before and after exposure to the stressor, hence the drop-in eye temperature could have been missed. The types of stressors also differed markedly among studies. Furthermore, when making comparisons between studies it is important to consider how the eye region was defined. Differences in this definition could result in the eye region comprising of different tissues and this could alter the temperature change observed in response to a stressor. However, it is important to note that many of the studies mentioned here used the same definition of the eye region [8, 9, 11, 12, 16, 19, 25] and opposing changes in eye temperature in response to a stressor were still observed.

In addition to the eye region, sympathetic-mediated cutaneous vasoconstriction stimulated by acute stress can be measured using other peripheral body regions. For example, IRT has also been used to detect changes in skin temperature in response to stress on the dorsal surface of deer [26], comb and wattle in poultry [14, 27] and nasal region in rhesus monkeys [28] and pigs [29]. Continuous change in temperature of the head region of sheep in response to an epinephrine infusion recorded in the present study (S1 Movie), suggests that other regions of the head, such as the ears, could potentially provide a more sensitive region to measure temperature changes in response to stress in sheep. However, it was beyond the scope of our study to measure temperature changes of other body regions.

Sheep and cattle appeared to react differently during the 5-min epinephrine infusion. In the present study, eye temperature tended to be higher and HRV numerically lower in sheep given epinephrine during the infusion period compared with sheep that received saline, but epinephrine had no effect on heart rate or norepinephrine concentrations. Conversely in cattle, eye temperature, heart rate and norepinephrine concentrations decreased and HRV increased during the epinephrine infusion period [19]. Lower RMSSD and SDNN suggests activation of the sympathetic nervous system in sheep in response to a 5-min epinephrine challenge, conversely, the increase in RMSSD and reduction in norepinephrine concentrations suggests that epinephrine altered the parasympathetic tone in cattle. An epinephrine challenge may have differentially altered the sympathetic/parasympathetic balance in sheep and cattle, leading to the difference in the HRV and eye temperature response between species. It would therefore be of interest to evaluate if the ANS response of sheep to more biologically relevant stressors, such as painful husbandry procedures or handling, would differ from cattle.

During the post-infusion period, heart rate, cortisol and PCV levels were higher and HRV was lower in sheep that received epinephrine, compared with sheep that received saline in the present study. These results are similar to those found in cattle in response to a 5-min epinephrine infusion challenge [19]. Similar to cattle, saline did not cause an increase in cortisol concentrations suggesting that handling and the infusion procedure did not stimulate the hypothalamic-pituitary adrenal response in the present study. The increase in cortisol and PCV in sheep suggests that epinephrine stimulated the sympathetic nervous system resulting in splenic contractions and activation of the hypothalamic-pituitary adrenal axis respectively. Moreover, lower RMSSD and SDNN during the post-infusion period further supports increased sympathetic tone in sheep in response to an epinephrine challenge.

Heart rate, HRV and PCV were measured in conjunction with eye temperature to validate that changes in eye temperature were associated with other measures of ANS activity in the

present study. The dose of epinephrine used was sufficient to cause a marked increase in plasma epinephrine concentrations as well as an increase in heart rate and PCV concentrations, and a reduction in HRV compared with sheep given saline. An increase in PCV is an indicator of splenic contractions during endogenous epinephrine release [30], and heart rate and HRV is influenced by both sympathetic and parasympathetic activity [2], therefore these results suggest that the epinephrine infusion was sufficient to activate the ANS in sheep. However, in the present study, there was only weak evidence that eye temperature increased in response to an epinephrine infusion which is in contrast to the marked change in eye temperature found in cattle in response to an epinephrine challenge. Our results are similar to Stubsjoen et al. [18] who reported that eye temperature, measured using IRT, did not change but SDNN and RMSSD tended to decrease in sheep in response to a noxious ischaemic stimulus. However, Cannas et al. [15] found that eye temperature in sheep was higher during restraint and a voluntary approach test. Once again, differences in data collection methodology between studies, such as timing of sampling, may account for these differences.

## Conclusions

Changes in HRV and eye temperature in response to an epinephrine infusion suggests, that in sheep, HRV is a more sensitive measure of ANS activity than eye temperature. However, further research is needed to confirm this finding using biologically relevant acute stressors. Only a small increase in eye temperature was found, however, further studies evaluating changes in temperature of other body regions, such as the ear, could be worthwhile to determine whether IRT would be a useful non-invasive method to measure stress and welfare in sheep.

## Supporting information

**S1 Movie. Example of a continuous infrared recording.** Movie shows a continuous infrared recording collected at a rate ~60 frames/s and a resolution of 640 x 480 pixels of the head of a sheep given an epinephrine infusion. White pixels represent temperatures exceeding 38.9˚C. (MOV)

**S1 Dataset. Dataset of study evaluating the effect of an epinephrine challenge on the autonomic response in sheep.**
(XLSX)

## Acknowledgments

Technical assistance from AgResearch staff was greatly appreciated, in particular Martin Upsdell for his assistance with the Flexi analysis.

## Author Contributions

**Conceptualization:** Mhairi A. Sutherland, Gemma M. Worth, Suzanne K. Dowling, Vanessa M. Cave, Mairi Stewart.

**Data curation:** Gemma M. Worth, Suzanne K. Dowling, Gemma L. Lowe, Mairi Stewart.

**Formal analysis:** Vanessa M. Cave.

**Funding acquisition:** Mairi Stewart.

**Investigation:** Gemma M. Worth, Suzanne K. Dowling, Gemma L. Lowe, Mairi Stewart.

**Methodology:** Mhairi A. Sutherland, Gemma M. Worth, Suzanne K. Dowling, Gemma L. Lowe, Mairi Stewart.

**Project administration:** Gemma M. Worth, Suzanne K. Dowling, Mairi Stewart.

**Supervision:** Suzanne K. Dowling, Mairi Stewart.

**Visualization:** Suzanne K. Dowling, Mairi Stewart.

**Writing – original draft:** Mhairi A. Sutherland, Gemma L. Lowe, Mairi Stewart.

**Writing – review & editing:** Mhairi A. Sutherland, Gemma M. Worth, Suzanne K. Dowling, Gemma L. Lowe, Vanessa M. Cave, Mairi Stewart.

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
