## [Decision Letter · Decision Letter 0]

12 Mar 2020

PONE-D-19-33285

Evaluation of infrared thermography to measure the autonomic nervous response in sheep­­­­

PLOS ONE

Dear Dr. Sutherland,

Thank you for submitting your manuscript to PLOS ONE. After careful consideration, we feel that it has merit but does not fully meet PLOS ONE’s publication criteria as it currently stands. Therefore, we invite you to submit a revised version of the manuscript that addresses the points raised during the review process.

You will find detailed comments from the two independent reviewers below. Please take them into consideration when revising the paper. I would like to draw special attention to the detailed comments by Reviewer 1.

In addition, based on my own reading of the manuscript, I ask that you revise the discussion so that it focuses primarily on the results of your study and how they relate to what has been published previously. As both reviewers point out, there is presently too much overlap with the literature review in the introduction section, and parts of the discussion are not well linked to your own results. 

A minor detail

Lines 73-74 I think you intend to say "Moreover, eye temperature has commonly been measured in many species because this area is not affected by the presence of hair"

We would appreciate receiving your revised manuscript by Apr 26 2020 11:59PM. To enhance the reproducibility of your results, we recommend that if applicable you deposit your laboratory protocols in protocols.io, where a protocol can be assigned its own identifier (DOI) such that it can be cited independently in the future. For instructions see: http://journals.plos.org/plosone/s/submission-guidelines#loc-laboratory-protocols

We look forward to receiving your revised manuscript.

Kind regards,

I Anna S Olsson, Ph.D.

Academic Editor

PLOS ONE

Journal Requirements:

2) Thank you for stating the following in the Financial Disclosure section:

[This study was funded by the New Zealand Foundation for Research, Science and

Technology (C10X0813). The funders had no role in study design, data collection and

analysis, decision to publish, or preparation of the manuscript.].   

We note that one or more of the authors are employed by a commercial company: AgResearch Ltd.

i. Please provide an amended Funding Statement declaring this commercial affiliation, as well as a statement regarding the Role of Funders in your study. If the funding organization did not play a role in the study design, data collection and analysis, decision to publish, or preparation of the manuscript and only provided financial support in the form of authors' salaries and/or research materials, please review your statements relating to the author contributions, and ensure you have specifically and accurately indicated the role(s) that these authors had in your study. You can update author roles in the Author Contributions section of the online submission form.

ii. Please also provide an updated Competing Interests Statement declaring this commercial affiliation along with any other relevant declarations relating to employment, consultancy, patents, products in development, or marketed products, etc. 

Reviewers' comments:

Reviewer's Responses to Questions

**Comments to the Author**

1. Is the manuscript technically sound, and do the data support the conclusions?

Reviewer #1: Yes

Reviewer #2: Partly

2. Has the statistical analysis been performed appropriately and rigorously? 

Reviewer #1: Yes

Reviewer #2: No

3. Have the authors made all data underlying the findings in their manuscript fully available?

Reviewer #1: Yes

Reviewer #2: Yes

4. Is the manuscript presented in an intelligible fashion and written in standard English?

Reviewer #1: Yes

Reviewer #2: No

5. Review Comments to the Author

Reviewer #1: This is an interesting paper investigating changes in eye region temperature as captured with IRT. The work is sound and adds to the growing literature on the potential value of surface temperature measures in stress assessment. However the authors must provide a stronger justification for the use of an epinephrine infusion rather than a relevant stressor. I also have various questions about methodology. In particular I would like reassurance on the effectiveness of habituation and a better description of the experimental procedure with regard to other possible sources of stress.

Introduction

The relevant literature is nicely reviewed, but I expect a stronger explanation for the rationale of using epinephrine infusion rather than a biologically relevant acute stressor. Almost all other studies in this field have used relevant stressful events to examine surface temperature responses – why was this approach not taken? I suspect there was some activation of stress due to experimental procedures in any case (see comments below) but this was not the aim.

Methods

The experimental procedures are not at clear in several places and more detail is needed. Specifically:

What handling was required to administer the treatments and collect blood samples?

What evidence (behavioural, HR, hormonal?) was there that habitation to handling and restraint was effective prior to the experiment starting? Otherwise possible influences on the stress response must be expected and fully discussed.

The total duration of thermal recording is not provided – was this as for blood sampling or for longer? Where were the animals during this time? Were they restrained? How was distance from the camera managed if the animals were not restrained? The description ‘camera positioned at a distance of approximately 1 m’ is problematic, as even small distance changes can affect accuracy – what was the range?

The IRT variable might be more accurately referred to as ‘eye region temperature’ since structures other than the eye itself were included.

Again what evidence is there that habituation to wearing HR monitors was successful? What was the duration of habituation? I ask these questions because for a study on acute stress it is crucial to know to what extent the experimental procedure affected the phenomenon under study.

Movie – this is useful but consider rescaling the temperature bar to the right – areas around the eye appear as white which is not in the calibration bar (presumably in excess of 38 degrees C).

Results

A mixture of P= and P< is reported – suggest consistency with P= except for P<0.001. This applies to the abstract as well.

Why are there so few sampling points for cortisol? These are so few as to be rather uninformative. The upward trajectory of the epinephrine treatment line suggests that further increase was possible, particularly given the fact that 15 mins is quite a short timescale for a cortisol response – this should be acknowledged.

Given that eye region temperature change is the focus of the paper, it would be good to see some more detailed data rather than a single figure. At least provide mean and SD temperature increase and range – this would allow more ready comparison with other work. Currently this information is difficult/impossible to glean accurately from the figure.

Can the authors report which region of the eye was usually the position of their maximum temperature recording? This may be relevant to interpretation (see below).

Discussion

In the discussion of why some studies show eye temp increases in response to stress while others show decreases, the authors mention differences in timing of recoding (which is fair enough and it is true that the initial drop can be very rapid and so missed) but it is also important to mention that different studies use different definitions of ‘eye’ when reporting temperature change, such that some are imaging the cornea, while others are imaging the whole eye region (and taking maximums from various sub regions thereof). These tissues differ markedly in various characteristics including peripheral circulation and its control by the ANS – this should be reflected in the authors discussion.

The authors notice that body surface areas other than the eye may reveal SIH – but say this is beyond the scope of the current study. Fair enough, but this seems a shame since at least one ear seems to be visible at all times in the movie and surface temperature analysis would be feasible.

The authors conclude that the HPA axis was not stimulated, but based on the figure I think care should be taken here – see comment above that the upward trajectory of the epinephrine treatment line suggests that further increase was possible. It should be acknowledged that 15 mins is a short timescale for a cortisol response which may well have continued/appeared beyond this time.

Lines 345-362 – many relevant factors for effective IRT data collection are discussed, but without adequate reference to the current study – do any of these limitations apply or if not how were they overcome and how do we know that? As it is it is rather generic and does not add much to the discussion.

Conclusion

Suggest rewording to “Changes in HRV and eye temperature in response to an epinephrine infusion suggests that, in sheep, HRV is a more sensitive non-invasive method than IRT to measure ANS activity (as induced by epinephrine infusion).” – since we don’t know if the same results would have been seen had a biologically relevant acute stressor been applied.

This point should also be indicated in the abstract.

Reviewer #2: This manuscript approaches an interesting scientific question to the assessment of autonomic nervous response with non-invasive methods. Considering the conclusion “These results suggest that heart rate variability is a sensitive, non-invasive method that can be used to measure ANS activity in sheep, whereas change in eye temperature measured using IRT is a less sensitive method”, authors may consider a change in focus, starting from the title, and put more emphasis to heart rate variability throughout the paper. If temperature remains as part of the title and the objective, authors should give its numerical values in the abstract and in the results section in a more explicit manner. It is important to improve the introduction, to better present to readers the relevance of this work. Please provide an explicit and complete statement of the objective of the work in the end of the introduction. There is excessive repetition of text between introduction and discussion.

A major issue with the manuscript is the ambiguity on the effects of treatment on eye temperature, as detailed below. If authors have the thermographic images of the head, they could verify the hypothesis they bring that other regions, such as ears, may be more responsive in terms of superficial temperature. Additionally, the discussion needs considerable improvement through a more extensive review of the literature and increased complexity.

Specific Comments

L81: Please refine link to previous paragraph, which presented a list of stressful situations.

L83: Please explain kind of noxious stimulus and animal species involved, to improve the flow of reading.

L219-220: what are the numeric values?

L245, 257: Do you mean “affected”?

L304-305: Do you have the data on temperature on other body regions? It could significantly improve this manuscript.

L274: Was it only a tendency for temperature increase in this study? But how come in the abstract it is stated that there was a difference (P<0.05, L31-32)? Then again in the conclusion it is understandable in the first phrase that there were changes in the eye temperature (L364).

6. PLOS authors have the option to publish the peer review history of their article (what does this mean?). If published, this will include your full peer review and any attached files.

Reviewer #1: No

Reviewer #2: No

---

## [Author Response · Author response to Decision Letter 0]

22 Apr 2020

The authors would like to thank the editor and the reviewers for their thoughtful and very helpful comments on this manuscript. The authors have considered all the comments listed below, have responded to these comments individually and made the appropriate changes to the manuscript. 

Editor

I ask that you revise the discussion so that it focuses primarily on the results of your study and how they relate to what has been published previously. As both reviewers point out, there is presently too much overlap with the literature review in the introduction section, and parts of the discussion are not well linked to your own results.

Au: Based on your comments the discussion is now more focused on the results of our study and how they relate to previous studies. We have reduced the level or repetition between the introduction and discussion.

Lines 73-74 I think you intend to say "Moreover, eye temperature has commonly been measured in many species because this area is not affected by the presence of hair"

AU: This sentence has now been changed to ‘Temperature of the eye region has commonly been measured in many species because this area is not affected by the presence of hair’.

Requested removal of “an anatomical region”

AU: Changed as requested. 

REVIEWER 1

Expects stronger explanation for the rationale for using epinephrine infusion rather than a biologically relevant acute stressor. Almost all studies in the field have used relevant stressful events to examine surface temp responses- why was this approach not taken? 

AU: This study was designed as a validation study so we wanted to make sure that we activated the ANS hence we used an epinephrine infusion. A similar study design was successfully used in cattle to validate IRT as a non-invasive measure of the ANS. As this was a validation study the purpose was not to measure how an animal responds to a particular biological stressor. However, this is an important next step and we have mentioned this in the discussion. Also, we have clarified the explanation of the objectives in the introduction.

I suspect there was some activation of stress due to experimental procedures in any case but this was not the aim.

AU: Yes, it is very likely that there was some activation of the stress response in these sheep, but to help reduce this we habituated the sheep to the facilities and equipment for 2 weeks before the start of the study. We also included a control group (saline), to take into account any changes in response to stress that was not associated with the epinephrine infusion.

What evidence (behavioural, HR or hormonal?) was there that habituation to handling and restraint was effective prior to the experiment starting? Otherwise possible influences on the stress response must be expected and fully discussed.

Again what evidence is there that habituation to wearing HR monitors was successful? What was the duration of habituation? I ask these questions because for a study on acute stress it is crucial to know to what extent the experimental procedure affected the phenomenon under study.

AU: Sheep were acclimated to the research facility and equipment for 2 weeks prior to the start of the study. This information has been clarified in the methods section. We would expect after this period of time that all animals would be habituated, and they did appear so, however we took no measures to prove this. This has been clarified in the methods section. We also had a control group in this study that only received saline to take into account for any ANS changes not related to the epinephrine.

What handling was required to administer treatments and collect blood samples?

AU: Animals were manually restrained to put the heart rate monitors on and then strained in the chute for the other measures. This has been clarified in the methods section.

The total duration of the thermal imaging is not provided- was this as for blood sampling or longer? Where were the animals during this time? Were they restrained? How was camera distance managed if the animals were not restrained?

AU: Within this section we have clarified the duration of the infrared recordings as being 25 mins both before and following the start of the infusion period. We have also explained that infrared recordings were collected while the animal was restrained in the chute and have clarified how distance and angle were kept consistent. 

The description “camera positioned at a distance of approximately 1m” is problematic, as even small distance changes can affect accuracy- what was the range?

AU: The word “approximately” has been removed as the distance and angle of the camera from the animal was keep constant throughout the duration of the study by the infrared camera being used with a tripod that was placed at a marked location 1m away from the animal. This has been clarified in the methods section. 

This is useful but consider rescaling the temperature bar to the right- areas around the eye appear as white which is not in the calibration bar (presumably in excess of 38 C) 

AU: We do not have the ability to rescale the temperature bar in this example video. However, for clarity we have provided an explanation in the figure legend that the white pixels represent those in excess of 38.9C. 

A mixture of P= and P< is reported- suggest consistency with P= except for P<0.001. this applies to the abstract as well.

AU: The significance of the treatment by time interaction (treatment x time) was assessed using an F-test and the exact p-values have been included in the text. The unprotected Fisher's least significant differences at the 5% level were used to compare the treatment means at each time period, hence for these comparisons we only have p-values of < or > 0.05. 

Why are there so few sampling points for cortisol? These are so few as to be rather uninformative. The upward trajectory of the epinephrine treatment line suggests that further increase was possible, particularly given the fact that 15 mins is quite a short timescale for a cortisol response- this should be acknowledged.

AU: Yes, the cortisol response to epinephrine may have increased further after 15 minutes, however, blood samples were only collected up to 15 minutes after the start the infusion. This is because we were predominantly interested in the acute ANS response (the 5-minute period during and after the epinephrine infusion) not the HPA response. 

Given that eye region temperature change is the focus of the paper, it would be good to see some more detailed data rather than a single figure. At least provide mean and SD temperature increase and range- this would allow more ready comparison to other work. Currently this information is difficult/impossible to glean accurately from the figure. 

AU: As requested we have included additional information e.g., mean and SD and ranges relating to the maximum eye temperature at the different stages of the experiment for the epinephrine and saline treatments. 

Can the authors report which region of the eye was usually the position of the maximum temperature recording? This may be relevant to interpretation.

AU: It is very important information and has now been added. The lacrimal caruncle was typically the position of the maximum eye temperature due to the rich capillary bed located at this region. This has been added to the manuscript. 

The authors conclude that the HPA axis was not stimulated, but based on the figure I think care should be taken here- see comment above that the upward trajectory of the epinephrine treatment line suggests that further increase was possible. It should be acknowledged that 15 mins is a short timescale for a cortisol response which may well have continued/appeared beyond this time. 

AU: We stated that HPA was not activated by saline (the control) but was activated by epinephrine ‘The increase in cortisol and PCV in sheep suggests that epinephrine stimulated the sympathetic nervous system resulting in splenic contractions and activation of the hypothalamic-pituitary adrenal axis respectively.” See responses above in regards to short time scale for cortisol measures.

In the discussion of why some studies show eye temp increases in response to stress while others show decreases, the authors mention differences in timing of recording (which is fair enough and it is true that the initial drop can be very rapid and so missed) but it is also important to mention that different studies use different definitions of “eye” when reporting temperature change, such that some are imaging the cornea, while others are imaging the whole eye region (and taking maximums from various sub regions thereof). These tissues differ markedly in various characteristics including peripheral circulation and its control by the ANS- this should be reflected in the authors discussion.

AU: We agree that region where eye temperature is measured is an important factor and have now added this point to the discussion.

The authors notice that body surface areas other than the eye reveal SIH- but say this is beyond the scope of the current study. Fair enough, but this seems a shame since at least one ear seems to be visible at all times in the movie and surface temperature would be feasible.

AU: It is a shame, but lack of funding has meant that it is beyond the scope of this study.

Many relevant factors for effective IRT data collection are discussed, but without reference to the current study- do any of these limitations apply or if not how were they overcome and how do we know that? As it is it is rather generic and does not add much to the discussion.

AU: The purpose of this paragraph was to highlight some of the considerations needed when measuring IRT, but we agree with the reviewer that it is not really relevant to our study as we took all these considerations into account when collecting the data. Therefore, this paragraph has been removed. 

Suggest rewording to “Changes in HRV and eye temperature in response to an epinephrine infusion suggests that in sheep, HRV is a more sensitive non-invasive method than IRT to measure ANS activity (as induced by epinephrine infusion).” Since we don’t know if the same results would have been seen had a biologically acute stressor been applied. 

AU: We agree with the reviewer, however, think it is redundant to state that this response was due to epinephrine twice in the same sentence. Therefore, we have added the sentence ‘However, further research is needed to confirm this finding using biologically relevant acute stressors’. 

REVIEWER 2:

Considering the conclusion 

“These results suggest that heart rate variability is a sensitive, non-invasive method that can be used to measure ANS activity in sheep, whereas change in eye temperature measured using IRT is a less sensitive method”

 Authors may consider a change in focus, starting from the title, and put more emphasis to heart rate variability throughout the paper. 

AU: The objective of our study was to evaluate if changes in eye temperature (measured using IRT) can be used to non-invasively measure ANS activity in sheep. Therefore, IRT was the focus of the study not HRV, so we have kept IRT the focus of the paper. HRV is already known to be a good measure of the ANS and was only included in the study to help validate IRT as a measure of the ANS. We have tried to clarify this in our objective in the introduction.

If temperature remains as part of the objective, authors should give its numerical values in the abstract and in the results section in a more explicit manner.

AU: The eye temperature data was analysed using Bayesian mixed model smoothing programme (refer to stats section). This model does not produce actual values, instead it determines the relationship between maximum eye temperature and treatment over time. Therefore, we can’t include numerical values here as they don’t relate to the statistical analysis. However, the descriptive means and standard deviations and ranges have been added to the results section.

It is important to improve the introduction to better present to readers the relevance of this work. There is excessive repetition between the intro & discussion

AU: We have amended the introduction to better highlight the focus of this work which was to validate IRT as a non-invasive measure of the ANS in sheep. Repetition between the introduction and discussion has been reduced.

Please refine link to previous paragraph, which presented a list of stressful situations.

AU: Link to following paragraph refined as requested.

Please explain kind of noxious stimulus and animal species involved, to improve the flow of reading.

AU: The ischaemia is the noxious stimulus as it causes pain. We have clarified this and also added that this particular study involved sheep. 

Please provide an explicit and complete statement of the objective of the work in the end of the introduction.

AU: We have provided a more explicit and complete statement of the study objectives at the end of the introduction.

What are the numeric values?

AU: The eye temperature data was analysed using Bayesian mixed model smoothing programme (refer to stats section). This model does not produce actual values, instead it determines the relationship between maximum eye temperature and treatment over time. However, the descriptive means and standard deviations and ranges have been added to the results section. 

Do you mean “affected”?

AU: Corrected 

Do you mean “affected”?

AU: Corrected 

Discussion needs considerable improvement through a more extensive review of the literature and increased complexity.

AU: We have focused on the literature which is relevant to what we measured in our study, hence we have focused on the IRT literature where eye temperature was measured in response to agricultural animals. It is unclear what increased complexity the reviewer is wanting. We feel that increased complexity often leads to increased speculation of the results and prefer not to do this. Instead we have tried to focus on the literature and findings that are relevant to our study. 

Was it only a tendency for temperature increase in this study? But how come in the abstract it is stated that there was a difference (P<0.005, L31-32)? Then again in the conclusion it is understandable in the first phrase that there were changes in the eye temperature (L364)

AU: When we used the term “tendency” we were meaning from a trajectory point of view rather than a statistical one, however, we have since removed “tendency” from the manuscript for clarity. 

A major issue with the manuscript is the ambiguity on the effects of treatment on eye temperature, as detailed below. If authors have thermographic images of the head, they could verify the hypothesis they bring that other regions, such as ears, may be more responsive in terms of superficial temperature. Do you have the data on temperature of other body regions? It could significantly improve this manuscript.

AU: We definitely agree that if we had this information we could answer this question about body region, however, as mentioned above it was beyond the scope of this study to measure temperature changes of other body regions. Therefore, unfortunately we do not have this additional data to include in the manuscript.

---

## [Editor Report · Decision Letter 1]

8 May 2020

Evaluation of infrared thermography as a non-invasive method of measuring the autonomic nervous response in sheep­­­­

PONE-D-19-33285R1

Dear Dr. Sutherland,

We are pleased to inform you that your manuscript has been judged scientifically suitable for publication and will be formally accepted for publication once it complies with all outstanding technical requirements.

With kind regards,

I Anna S Olsson, Ph.D.

Academic Editor

PLOS ONE
---

## [Editor Report · Acceptance letter]

14 May 2020

PONE-D-19-33285R1 

Evaluation of infrared thermography as a non-invasive method of measuring the autonomic nervous response in sheep­­­­ 

Dear Dr. Sutherland:

I am pleased to inform you that your manuscript has been deemed suitable for publication in PLOS ONE. Congratulations! Your manuscript is now with our production department. 

With kind regards,

on behalf of

Dr I Anna S Olsson 

Academic Editor

PLOS ONE